# Remote Sensing Image Fusion Based on Morphological Convolutional Neural Networks with Information Entropy for Optimal Scale

**DOI:** 10.3390/s22197339

**Published:** 2022-09-27

**Authors:** Bairu Jia, Jindong Xu, Haihua Xing, Peng Wu

**Affiliations:** 1School of Computer and Control Engineering, Yantai University, Yantai 264005, China; 2School of Information Science and Technology, Hainan Normal University, Haikou 571158, China; 3School of Information Science and Engineering, University of Jinan, Jinan 250024, China

**Keywords:** remote sensing image fusion, morphological component analysis, information entropy, deep learning, multi-scale

## Abstract

Remote sensing image fusion is a fundamental issue in the field of remote sensing. In this paper, we propose a remote sensing image fusion method based on optimal scale morphological convolutional neural networks (CNN) using the principle of entropy from information theory. We use an attentional CNN to fuse the optimal cartoon and texture components of the original images to obtain a high-resolution multispectral image. We obtain the cartoon and texture components using sparse decomposition-morphological component analysis (MCA) with an optimal threshold value determined by calculating the information entropy of the fused image. In the sparse decomposition process, the local discrete cosine transform dictionary and the curvelet transform dictionary compose the MCA dictionary. We sparsely decompose the original remote sensing images into a texture component and a cartoon component at an optimal scale using the information entropy to control the dictionary parameter. Experimental results show that the remote sensing image fusion method proposed in this paper can effectively retain the information of the original image, improve the spatial resolution and spectral fidelity, and provide a new idea for image fusion from the perspective of multi-morphological deep learning.

## 1. Introduction

Due to the limitations of satellite technology, most remote sensing images can only be panchromatic (PAN) images and low-resolution multispectral (LRMS) images of the same area. The goal of remote sensing image fusion is to fuse the spectral information of LRMS images and the spatial information of PAN images to generate a remote sensing image with both high spatial resolution and high spectral resolution [1]. Classical component substitution (CS) [2] methods are the most widely used, but they often result in spectral distortion. Multiresolution analysis (MRA) [3] methods are also often utilized. Compared with the CS method, methods based on MRA retain the spectral information better, but the spatial details are seriously lost. Model-based [4] methods have also been applied to remote sensing image fusion. The aforementioned methods can effectively reduce spectral distortion, but usually lead to blurred results.

The popular convolutional neural networks (CNN) method can learn the correlation between PAN images and LRMS images because of its excellent nonlinear expression and achieves better fusion results than traditional remote sensing image fusion methods [5,6]. Therefore, many existing fusion methods choose to combine traditional methods with deep learning methods [7,8,9] and have achieved good results. However, one of the basic tasks of image analysis and computer vision is to extract different features of an image. Most of the existing deep learning fusion methods treat the source image as a single component without considering the diversity of image components, thus ignoring the different morphological details in the source image. Remote sensing image usually contain spectral information and spatial structure, among which the PAN image reflects the spatial distribution information and structure information of the image. The texture component of the PAN image contains the image surface information and its relationship with the surrounding environment, which can better reflect the spatial structure information of the PAN image. The boundary of the cartoon component of remote sensing image is smoother and the spectral information is retained, so the spectral information of the LRMS image can be completely characterized by its cartoon component, and the redundancy and noise can be filtered out.

Morphological component analysis (MCA), proposed by J. Starck et al. [10,11], has been used to solve problems such as image decomposition [12], image denoising [13], and image restoration [14]. The main idea of this algorithm is to associate each morphological component in the data with a dictionary of atoms. Each component of the image is assumed to correspond to a suitable dictionary enabling the sparsest representation vector. The sparse vector is reconstructed according to the corresponding dictionary to obtain the separated image components.

Therefore, in this paper, we propose a method combining the sparse decomposition-multi-scale MCA method and CNN for remote sensing image fusion, with optimal scale determined by information entropy. We use MCA to sparsely decompose the original images and acquire the texture components and cartoon components at multi-scale. Considering the variability of the different components of the image, we use information entropy to calculate the threshold of the decomposition parameters. This facilitates the extraction of the different components at the optimal scale and effectively acquires more detail from the image. We use the spectral and spatial information of the LRMS and PAN images, respectively, to input the cartoon component of the LRMS remote sensing image and the texture component of the PAN image into an attentional CNN for fusion. The remainder of this paper is organized as follows. Section 2 describes the multi-scale MCA method. Section 3 details the fusion network and displays multi-scale fusion results. Section 4 provides the overall experimental results and analysis. Finally, Section 5 concludes this research.

## 2. Multi-Scale MCA Algorithm

### 2.1. Image Decomposition via MCA

We represent an image as f=u+v, where u is the cartoon component of f, which is smooth and contains the geometric feature information of the image. v represents the texture component of the image and is the high-frequency part of the image. Decomposing an image into cartoon and texture components is essential for many applications. MCA joins two transform bases to sparsely decompose the image, and the joint local discrete cosine transform (LDCT) and curvelet transform (CT) are used as MCA decomposition dictionary: $D=\left[D_{1},D_{2}\right]$. This enables the extraction of the texture components and cartoon components of the image, where $D_{1}$ represents the LDCT dictionary and $D_{2}$ represents the CT dictionary. 

Assuming that the remote sensing image contains only the texture component $X^{T}$, the LDCT dictionary $D_{1}$ can sparsely represent the texture image. The Equation for solving the texture sparse coefficient is as follows:(1)$$\alpha_{PAN}^{T}=\underset{\alpha^{T}}{\text{Arg min}}\|\alpha^{T}{\|}_{0}\;\;\;\;\;\;\;\;\;\;\;\;\;\;\;subject\;to:X^{T}=D_{1}\alpha^{T}$$
where $\|u{\|}_{0}$ denotes the $l^{0\;}$ norm that effectively calculates the number of non-zero entries in the vector $X^{T}$ and $\alpha^{T}$ is the coefficient for the dictionary representation. The LDCT dictionary $D_{1}$ represents the non-texture components in the image as zeros, maximizing the sparseness. The dictionary $D_{1}$ is sparse with respect to the texture components of the image but not sparse to the cartoon components of the image. Thus, the texture components of the remote sensing image are obtained using the above model.

Similarly, for a remote sensing image $X^{C}$ that contains only cartoon components, the image is represented by the CT dictionary $D_{2}$, which is sparse only with respect to cartoon components. The equation is as follows:(2)$$\alpha_{MS}^{C}=\underset{\alpha^{C}}{\mathrm{Arg}\;\min}\|\alpha^{C}{\|}_{0}\;\;\;\;\;\;\;\;\;\;\;\;\;\;\;subject\;to:X^{C}=D_{2}\alpha^{C}$$
where $\alpha^{C}$ is the coefficient for the dictionary. Using the CT dictionary $D_{2}$, the non-cartoon elements in the image are represented as zeros. Because the CT dictionary only represents sparse cartoon components, this model extracts the cartoon components in a remote sensing image.

According to the above model, for any remote sensing image X containing both texture and cartoon components, it is necessary to decompose the components with the joint decomposition dictionary D containing both dictionary $D_{1}$ and dictionary $D_{2}$, posing the following regularization problem:(3)$$\left\{\alpha_{PAN}^{T},\alpha_{MS}^{C}\right\}=\underset{\left\{\alpha^{T},\alpha^{C}\right\}}{\mathrm{Arg}\;\min}\|\alpha^{T}{\|}_{0}+\|\alpha^{C}{\|}_{0}\;\;\;\;\;\;\;\;\;\;subject\;to:X=D_{1}\alpha^{T}+D_{2}\alpha^{C}$$

To better retain fused image information, we analysis the morphological components of the PAN image with a single channel and the MS image with three channels, obtaining the texture components of the PAN image and cartoon components of the MS image. Equations (4) and (5) show the sparse decomposition of the PAN image and MS image, respectively:(4)$$X_{PAN}=X_{PAN}^{T}+X_{PAN}^{C}=D_{1}\alpha_{PAN}^{T}+D_{2}\alpha_{PAN}^{C}=D\left(\alpha_{PAN}^{T}+\alpha_{PAN}^{C}\right)$$
(5)$$X_{MS}=X_{MS}^{T}+X_{MS}^{C}=D_{1}\alpha_{MS}^{T}+D_{2}\alpha_{MS}^{C}=D\left(\alpha_{MS}^{T}+\alpha_{MS}^{C}\right)$$
where $\alpha_{PAN}^{T}$, $\alpha_{PAN}^{C}$, $\alpha_{MS}^{T}$, and $\alpha_{MS}^{C}$ represent the corresponding decomposition coefficients. $X_{PAN}^{T}$ and $X_{PAN}^{C}$ are texture and cartoon components of the PAN image, respectively. $X_{MS}^{T}$ and $X_{MS}^{C}$ are texture and cartoon components of the MS image, respectively.

### 2.2. Decomposition with Different Scales

The existing MCA method uses a single scale [15], while humans analyze remote sensing images with complex components at multi-scale. This inspires the analysis of the image at multi-scale for morphological components, and the decomposition of the remote sensing image into texture and cartoon components at multi-scale.

Different MCA decomposition parameters represent different scales, and different scales also represent different resolutions. As shown in Figure 1 and Figure 2, we decompose the MS and PAN images into cartoon and texture components at different scales, and we set five decomposition parameters with 16/512, 32/512, 64/512, 128/512, and 256/512. Figure 1 and Figure 2 show that the cartoon component of the MS image and the texture component of the PAN image are decomposed at different scales (resolutions) with different parameters.

As shown in Figure 1 and Figure 2, the image components at different scales are not the same. Figure 1 indicates that a small threshold value removes too many edge details from the MS image, resulting in side effects such as noise, ultimately causing spectral distortion of the fused image. Figure 2 indicates that a large threshold value removes too many texture details from the PAN image, resulting in insufficient component information, ultimately causing noise in the fused image. Our target is to preserve details, remove redundant information and noise, and effectively retain texture and cartoon components. Therefore, controlling the parameter thresholds to construct a multi-scale dictionary is essential to achieve sparse multi-scale component decomposition.

### 2.3. Information Entropy Metric

Information entropy reflects the amount of information contained in an image at a certain position [16,17]. The threshold value of the control parameter is calculated using information entropy to retain the rich amount of information contained in the image while eliminating irrelevant information. This facilitates morphological component decomposition at multi-scale and selects the fusion results at the optimal scale.

In our previous work [18], we assume that T and C are the two images to be fused, the joint information entropy of the fused images can be expressed as $H\left(T,C\right)$. The conditional information entropy can be expressed as $H\left(T/C\right)$ and $H\left(C/T\right)$, and the mutual information entropy is $M\left(T;C\right)$, representing the redundant information (repeated content) between T and C. Then, the relationship between them can be expressed as Equation (6) [19]. The relationship between the information entropy of the two input source images is also described in Figure 3.
(6)$$H\left(T,C\right)\;=H\left(\frac{T}{C}\right)+H\left(\frac{C}{T}\right)+M\left(T;C\right)$$

The ideal fusion goal of image T and image C is that the information entropy of the fused image is $H\left(T,C\right)$. However, in the actual fusion process, in addition to the redundant information $M\left(T;C\right)$, other noise and interference may also exist, affecting the fusion results. Figure 4 expresses the relationship between noisy image T and noisy image C. Thus, considering noise, the remote sensing image fusion process ideally maintains the maximum joint information entropy of the input source image.

Based on the above analysis, assuming that $F\subseteq R^{N\times N}$ represents the fused image of size N×N pixels, we first average the RGB values of the three channels in the same pixel position and convert the color image into a gray image. Then, the image is classified into L gray levels. $f_{i}$ denotes the gray value of the pixel with spatial index i in the image, where $f_{i}\in G_{L}=\left\{0,1,\ldots ,L-1\right\}$. Based on the theory of information entropy, $\tilde{f_{i}}$ is the mean gray value over the neighborhood of the fused image. The neighborhood mean gray value composes the spatial feature vector of the gray distribution and can form a feature binary group with the pixel gray values of the image ($f_{i},\tilde{f_{i}}$). The comprehensive feature $X_{f_{i},\tilde{f_{i}}}$ of the gray value and the gray distribution of surrounding pixels is expressed as:(7)$$X_{f_{i},\tilde{f_{i}}}=\frac{g\left(f_{i},\tilde{f_{i}}\right)}{N^{2}}$$
where $g\left(f_{i},\tilde{f_{i}}\right)$ represents the number of occurrences of a single pixel feature binary group at a certain position. Combined with the two-dimensional information entropy of the image, Equation (8) calculates the entropy value of the final fused image F.
(8)$$H_{F}=-\underset{i=0}{\sum^{L-1}}X_{f_{i},\tilde{f_{i}}}\log X_{f_{i},\tilde{f_{i}}}$$

The information entropy $H_{F}$ of the image at different fusion scales is calculated by Equation (8) to gain the amount of the information of the fused image and utilize to determine the optimal fusion threshold.

### 2.4. Multi-Scale Spatial Attention Module

Selective visual attention enables humans to quickly locate salient objects in complex visual scenes, inspiring the development of algorithms based on human attention mechanisms [20]. In the field of deep learning, the attention mechanism can be seen as a weighted combination of input feature mappings, where the weights depend on the similarity between the input elements. Spatial attention is used to determine the location salient information in a target image. For the remote sensing image with complex structures, the lack of spatial structure leads to inaccurate positioning, with different weights between different regions of the same channel. Spatial attention is calculated using Equation (9).
(9)$$M_{S}\left(F\right)=\sigma \left\{f^{5\times 5}\left\{\left[AvePool\left(F\right);MaxPool\left(F\right)\right]\right\}\right\}=\sigma \left\{f^{5\times 5}\left\{\left[F_{avg}^{s};F_{max}^{s}\right]\right\}\right\}$$
where σ denotes the sigmoid activation function, F denotes the feature map, and $AvePool\left(\cdot \right)$ and $MaxPool\left(\cdot \right)$ denote average pooling and maximum pooling, respectively. $f^{5\times 5}$ denotes a convolution operation with a 5×5 pixel kernel. In this paper, we add a spatial attention module under each scale to enhance the information interaction in space and to strengthen the focus on valid information along the spatial dimension. The structure of multi-scale spatial attention is denoted by the dotted box in Figure 5b.

## 3. Methods

The proposed method is mainly composed of three parts, including MCA, feature extraction and feature fusion respectively. Firstly, the PAN image and the MS image are decomposed by MCA, the multi-scale texture components of PAN image and the multi-scale cartoon components of LRMS image are obtained. The spectral and spatial information are preserved while the redundancy and noise are removed. As shown in Figure 5a, the feature extraction network module is composed of two branches cascade convolution layers, which extract spectral features and spatial features obtained by MCA, respectively. Then, feature fusion network is used to generate the MS image with high spatial resolution. Finally, the optimal fusion scale is judged by information entropy theory, so as to get the high-resolution multispectral (HRMS) image under the optimal scale.

### 3.1. Network Setup and Multi-Scale Fusion

#### 3.1.1. Algorithm Flow

Figure 6 shows the flow chart of the proposed fusion method; the detailed steps are as follows:A The joint LDCT dictionary $D_{1}$ and CT dictionary $D_{2}$ form the decomposition dictionary $D=\left[D_{1},D_{2}\right]$, and MCA is performed on the input source images PAN and MS at multi-scale to extract the texture component and cartoon components, respectively.The threshold values of the parameters are calculated using the information entropy of the fused image from Step 3 to select the best extraction scale for the cartoon component $X_{MS}^{C}$ of the MS image and the texture component $X_{PAN}^{T}$ of the PAN image.The optimal-scale cartoon component $X_{MS}^{C}$ and texture component $X_{PAN}^{T}$ are fused by the attentional CNN to produce the final fused image.

#### 3.1.2. Network Structure

$PAN\left(i,\;j\right)$ and $MS\left(i,\;j\right)$ are the corresponding pixels of the PAN image and MS image at position $\left(i,\;j\right)$, respectively. $T\left(i,\;j\right)$ and $C\left(i,j\right)$ are the pixels at the corresponding points of the texture component and the cartoon component, respectively. The fused image F is obtained by calculating the fused pixels $F\left(i,\;j\right)$. Let $NT\left(i,\;j\right)$ and $NC\left(i,\;j\right)$ be the neighboring pixel points of $T\left(i,\;j\right)$ and $C\left(i,\;j\right)$, respectively. The texture component and cartoon component are through a 3×3 pixel convolution kernel to calculate NT and NC, respectively. Then, these neighboring pixels pass through a 1×1 pixel convolution kernel to obtain the fused image F.

Figure 5 shows the overall network model. The entire fusion network comprises 10 convolutional layers, where six convolutional layers $X_{Fusioni}^{T}$ and $X_{Fusioni}^{C}$ (i=1, 2, 3) have convolutional kernels of size 3×3 pixels and the remaining convolutional layers have convolutional kernels of size 1×1 pixel. After each linear convolution operation, we incorporate the Leaky ReLU (LReLU) activation function to further improve the fused image. The convolution operations are expressed in Equation (10).
(10)$$F=\mathrm{LReLU}\left(X*w\right)$$
where X represents the input to the convolution. w is the convolution kernel and $\mathrm{LReLU}\left(X\right)=\max\left\{0,x\right\}$ is the nonlinear activation function.

In the fusion network, $X_{Fusion1\;}^{TC}$ represents the fused image of the cartoon component $X_{Fusion}^{C}$ and the texture component $X_{Fusion}^{T}$ after weighted averaging. The computation process involves integrating the cartoon component and the texture component to construct the new image $X_{Fusion1}^{TC}$ and then applying the convolution operation. Unlike $X_{Fusion1}^{TC}$, the inputs of $X_{Fusion2}^{TC}$, $X_{Fusion3}^{TC}$, and $X_{Fusion4}^{TC}$ all contain three feature maps. For example, we obtain $X_{Fusion4}^{TC}$ by concatenating $X_{Fusion3}^{T}$, $X_{Fusion3}^{C}$, and $X_{Fusion3}^{TC}$ and then convolving them, where $X_{Fusion1}^{T}$, $X_{Fusion2}^{T}$, and $X_{Fusion3}^{T}$ have the same number of feature maps as $X_{Fusion1}^{C}$, $X_{Fusion2}^{C}$, and $X_{Fusion3}^{C}$ (32, 64, and 128, respectively). Similarly, $X_{Fusion1}^{TC}$, $X_{Fusion2}^{TC}$, and $X_{Fusion3}^{TC}$ have 32, 64, and 128 feature maps, respectively.

Let $T^{k}\left(k=1,2,3\right)$ and $C^{k}\left(k=1,2,3\right)$ denote the output of the kth convolutional layer in the two branches network of the texture component and the cartoon component, respectively. $T^{k}$ and $C^{k}$ are calculated using Equations (11) and (12), respectively.
(11)$$T^{k}\left(i,j,ch\right)=\mathrm{LReLU}\left(T^{k-1}\left(i,j,ch\right)*w_{T\left(ch\right)}^{k}\right)$$
(12)$$C^{k}\left(i,j,ch\right)=\mathrm{LReLU}\left(C^{k-1}\left(i,j,ch\right)*w_{C\left(ch\right)}^{k}\right)$$
where ch is the channel index. $w_{T\left(ch\right)}^{k}$ and $w_{C\left(ch\right)}^{k}$ denote the kth layer of convolution kernels for the texture component and the cartoon component, respectively.

In the fusion network, the fusion results of the previous layer are referred to in the convolution operation of each layer. For each pixel in the final fused image, we can choose to increase the size of its convolution kernel or use a deeper network model to expand the area of its corresponding pixel in the original image to improve the fusion ability of the network model.

#### 3.1.3. Different Scale Fusion Results

Section 2.3 details the selection of the parameter 128/512 as the optimal fusion scale of a set of remote sensing data. To corroborate that the fusion scale is optimal, we use the proposed attentional CNN to fuse the cartoon component extracted from the MS image and the texture component extracted from the PAN image at different scales, and Figure 7 shows the corresponding fusion results. The Figure 7 confirms that using the 128/512 decomposition parameter yields the fewest artifacts and superior fusion results. Furthermore, the information entropy diagram in Figure 8 also proves that the optimal fusion result is obtained by using the parameter 128/512. 

Figure 8 and Table 1 show the information entropy of the fused image F calculated by Equation (8). The details present in the fused results are different at different scales, and the calculation results show that the effective information contained in the image reaches saturation using the 128/512 scale parameters, and the spectral and spatial information of the source image is well preserved while removing part of the redundancy and noise. This is because small-scale decomposition parameters lose too much information from the original image, while an overly large scale does not increase the effective amount of information because of redundancy and noise.

## 4. Results and Discussion

### 4.1. Model Training

We use a regression model to train the fusion function: $fusion=F\left(PAN,MS\right)$, using the $l_{2}$ paradigm as the loss function, as expressed in Equation (13).
(13)$$L\left(\theta \right)=\frac{1}{n}\underset{i=1}{\sum^{n}}\|I-Fusion\left(\theta ;PAN,MS\right){\|}^{2}$$
where I is the original image from the training set, PAN represents a PAN image, and MS is a low-resolution multispectral image. $Fusion\left(\theta ;PAN,MS\right)$ is the fusion function of the model output and the number of training samples is denoted by *n*. To solve the fusion function Fusion, we need to minimize the I. The pixel values of the image range from 0–255 and are normalized to the interval $\left[0,\;1\right]$ before being input to the model.

Adam’s algorithm [21], an adaptive learning rate optimization algorithm of stochastic gradient descent, is used as the optimization algorithm of our model. The initial learning rate of the model was set to 0.001 and divided by 10 at 50% and 75% of the total number of training phases. The training took 50 min per cycle and we trained for eight cycles. The final training mean squared deviation of the model was 0.00017.

### 4.2. Experimental Data

To assess the effectiveness of the proposed method, we conducted experiments on four sets of remote sensing images with different topographical areas. The first set of experimental data (Figure 9a,b) is obtained by the SPOT-6 satellite, which captures PAN images with a spatial resolution of 1.5 m and MS images with a spatial resolution of 6 m. Figure 10 shows the histogram of the evaluation indexes of each experimental result of the first set of experimental data. The second set of experimental data (Figure 11a,b) is obtained by the WorldView-2 satellite, which captures PAN images with a spatial resolution of 0.5 m and MS images with a spatial resolution of 2 m. Figure 12 shows the histogram of the evaluation indexes of each experimental result of the second set of experimental data. The third set of experimental data (Figure 13a,b) are MS images with a resolution of 19.5 m from the China-Brazil Earth Resources Satellite (CBERS) image and PAN images with a resolution of 15 m from the Landsat ETM+ image. The test area is located in Doumen District, Zhuhai City, Guangdong Province, including agricultural land, water bodies and forest land. Figure 14 shows the histogram of the evaluation indexes of each experimental result of the third set of experimental data The last set of experimental data (Figure 15a,b) are MS images with 4 m resolution and PAN images with 1 m resolution from IKONOS images. The experimental area is located in Beijing Normal University, and includes a playground, vegetation, and buildings. Figure 16 shows the histogram of the evaluation indexes of each experimental result of the last set of experimental data

### 4.3. Evaluation Indexes

We use Figure 9a and Figure 11a as reference images to objectively verify the performance of different fusion methods in the first and second groups of experiments. We use four objective evaluation indexes to evaluate the experimental results: correlation coefficient (CC) [22], root mean square error (RMSE) [23], relative dimensionless global error synthesis (ERGAS) [22], and peak signal to noise ratio (PSNR) [24]. 

CC reflects the correlation between two images, and a larger correlation parameter indicates more similarity between two images.
(14)$$\mathrm{CC}\left(I_{H},I_{W}\right)=\frac{\sum_{i=1}^{M}\sum_{j=1}^{N}\left((I_{H}\left(i,j\right)-\bar{I_{H}}\right)\left(I_{W}\left(i,j\right)-\bar{I_{W}}\right)}{\sqrt{\sum_{i=1}^{M}\sum_{j=1}^{N}{\left(I_{H}\left(i,j\right)-\bar{I_{H}}\right)}^{2}\times \sum_{i=1}^{M}\sum_{j=1}^{N}{\left(I_{W}\left(i,j\right)-\bar{I_{W}}\right)}^{2}}}$$

Among them, $I_{H},I_{W}$ represent the pixels of the fused image and the ideal reference image respectively. $\bar{I_{H}},\bar{I_{W}}$ represent the average of pixels. The ideal CC value is 1. 

RMSE is the difference between the pixel values of the fused image and the reference image. The ideal value of RMSE is 0.
(15)$$\mathrm{RMSE}\left(I_{H},I_{W}\right)=\sqrt{\frac{1}{MN}\overset{M}{\underset{i=1}{\sum }}\overset{N}{\underset{j=1}{\sum }}(I_{H}\left(i,j\right)-{\left(I_{W}\left(i,j\right)\right)}^{2}}$$

The spectral and spatial quality of the fused image is evaluated using the ERGAS algorithm.
(16)$$\mathrm{ERGAS}=100\frac{h}{l}\sqrt{\frac{1}{L}\overset{L}{\underset{l=1}{\sum }}{\left(\frac{\mathrm{RMSE}\left(l\right)}{u\left(l\right)}\right)}^{2}}$$
where *h* and *l* represent the resolution of PAN image and MS image respectively. *L* is the number of bands. $u\left(l\right)$ is the mean value of the original MS band l. A smaller value indicates a higher quality fused image, and the ideal value is 0. 

PSNR reflects the degree of noise and distortion level of the image.
(17)$$\mathrm{PSNR}=10\times \log_{10}\left(\frac{{\left(2^{N}-1\right)}^{2}}{\frac{1}{MN}\sum_{i=1}^{M}\sum_{j=1}^{N}(I_{H}\left(i,j\right)-{\left(I_{W}\left(i,j\right)\right)}^{2}}\right)$$

The high value of PSNR indicates that the fused image is closer to the reference image and therefor of higher quality.

For the third and fourth groups of experiments, we use the following three common objective evaluation indexes to evaluate the experimental results: quality without reference (QNR) index [25], and two components $D_{\lambda }$ and $D_{s}$ to quantify the spectral distortion and spatial distortion, respectively [26].
(18)$$D_{\lambda }=\sqrt{\frac{2}{C\left(C-1\right)}\overset{C}{\underset{i=1}{\sum }}\overset{C}{\underset{j>1}{\sum }}\left|Q\left(\hat{I_{i},}\hat{I_{j}}\right)-Q\left(I_{i}^{LM},I_{j}^{LM}\right)\right|}$$
(19)$$D_{S}=\sqrt{\frac{1}{C}\overset{C}{\underset{i=1}{\sum }}\left|Q\left(\hat{I_{i},}P\right)-Q\left(I_{i}^{LM},P^{LM}\right)\right|}$$
where $I^{LM}$ represents the LRMS image and *C* represents the number of bands. $\hat{I}$ indicates the HRMS image, and *P* indicates the PAN image. Q denotes the Q-index.
(20)$$\mathrm{QNR}={\left(1-D_{\lambda }\right)}^{\alpha }{\left(1-D_{S}\right)}^{\beta }$$
where α and β are usually set to 1. The ideal value of QNR is 1, and the ideal value of $D_{\lambda }$ and $D_{s}$ is 0.

### 4.4. Experimental Results

The experimental results compare our proposed approach with Brovey [27], GS [28], IHS [29], ATWT [30], PCA [31], DWT [32], PanNet [33], FCNN [34], and PNN [35]. For our method, we use the calculated optimal fusion threshold to obtain the final experimental results. Figure 9, Figure 11, Figure 13, and Figure 15, respectively, show the experimental results of different satellite data.

**Figure 9 sensors-22-07339-f009:**
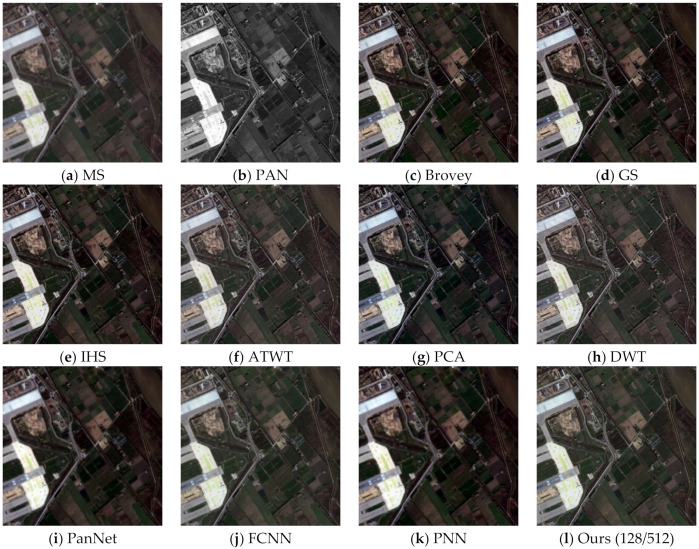
Fusion results for the first group of remote sensing image data.

Figure 9 shows the fusion results for the first set of data. As can be seen from Figure 9c–h, although the fusion images obtained by the traditional methods have high spatial resolution, the spectral color is too saturated and there is a large area of spectral distortion. Figure 9i,k show the fusion results of the two deep learning methods, with varying degrees of spectral distortion and low spatial resolution. The spectral distribution of landmarks and other parts in Figure 9j and the method in this paper (Figure 9l) are more uniform, and the color effect is closer to the spectral information of MS images. However, in comparison, our method better reflects the high-frequency detail features. In addition, in the wheat field and other large areas where the spectral information is relatively close, the effect of our method is optimal. Table 2 and Figure 10 display the evaluation indexes for the first set of data fusion results, where the bold numbers indicate the best score for each evaluation indexes. Compared with the other seven methods, our method achieves better results for all of the evaluation indexes. These quantitative results, in conjunction with the subjective visual results in Figure 9, show that our method outperforms existing fusion methods.

**Figure 10 sensors-22-07339-f010:**
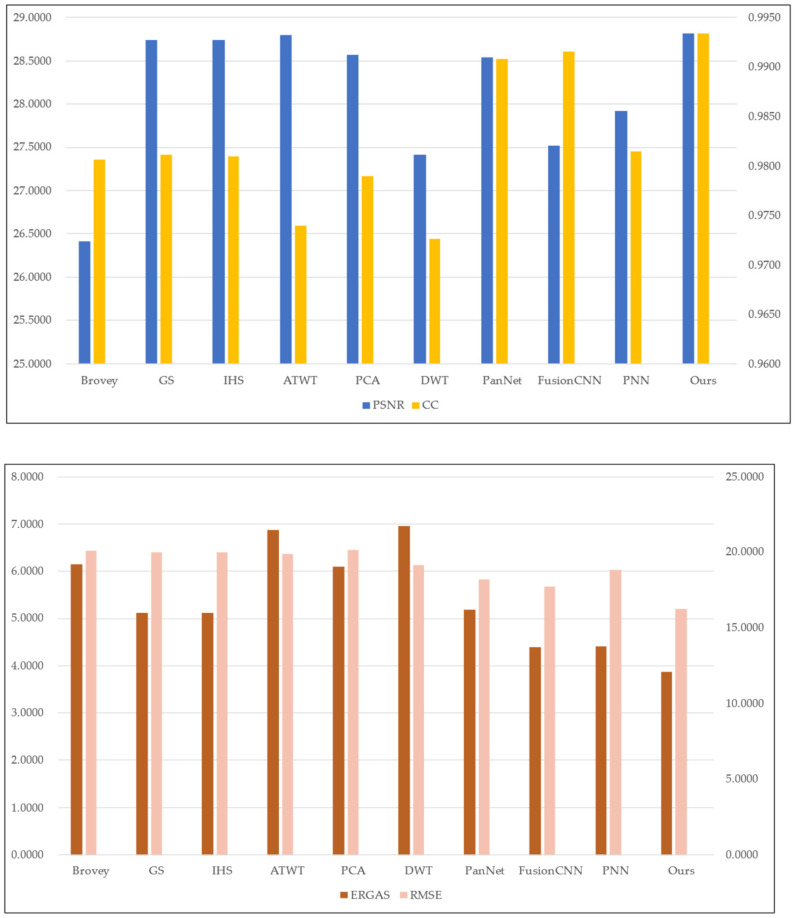
Histogram results of the first group of evaluation indexes.

Figure 11 shows the fusion results on the second set of data, which mainly contains mountains and vegetation. The traditional methods (Figure 11c,h) result in different degrees of distortion in vegetation color with over-brightness or darkness compared with the original MS image. Compared with the traditional methods, the deep learning methods used to obtain Figure 11i,k achieve better spectral quality but not high spatial quality. The spectral information of the vegetation part in Figure 11j does not reflect the obvious difference between light and dark, the edge of the mountain is not smooth enough, and the spatial resolution is not as good as that of the method in this paper (Figure 11l). These results combined with the evaluation indexes in Table 3 and Figure 12 show that our fusion results are superior.

**Figure 11 sensors-22-07339-f011:**
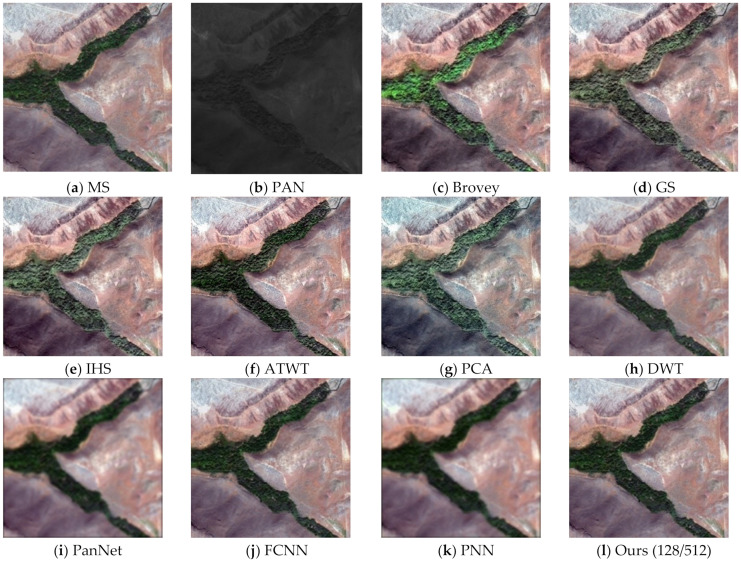
Fusion results for the second group of remote sensing image data.

**Figure 12 sensors-22-07339-f012:**
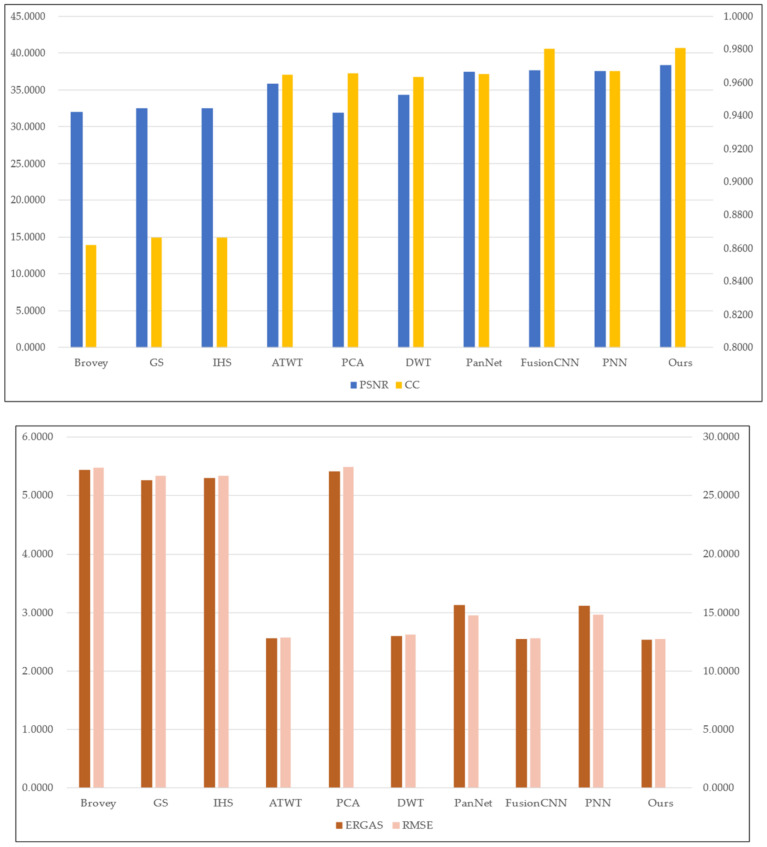
Histogram results of the second group of evaluation indexes.

Figure 13 shows the fusion results of different fusion methods on the third group of remote sensing images. Because of the relatively close resolution of images from this group of data sources, the optimal fusion scale is also different from the first two groups of experiments. All of the methods improve the quality of the fused images to some extent compared with the input PAN images and MS images. However, the fusion results of the traditional methods all show spectral distortion compared with the deep learning methods. It is clear from Figure 13c–h that both the mountainous part in the upper left corner and the vegetation part in the lower right corner exhibit more pronounced spectral distortion compared with the original multispectral image. Figure 13j and our method both retain the spectral and spatial information of the input image more completely, but our proposed fusion method still outperforms FCNN in terms of spatial information retention. Table 4 and Figure 14 list the third set of objective evaluation indexes. The bold numbers in Table 4 indicate the best value for each evaluation index. With the exception of the $D_{\lambda }$ metric, our method obtains the best results.

**Figure 13 sensors-22-07339-f013:**
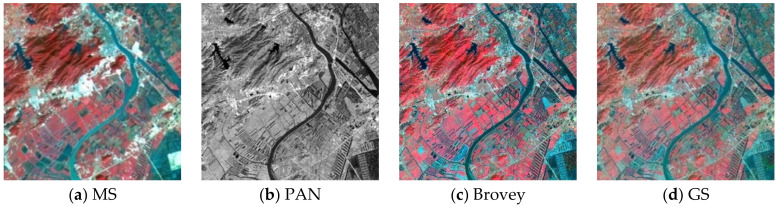
Fusion results for the third group of remote sensing image data.

**Figure 14 sensors-22-07339-f014:**
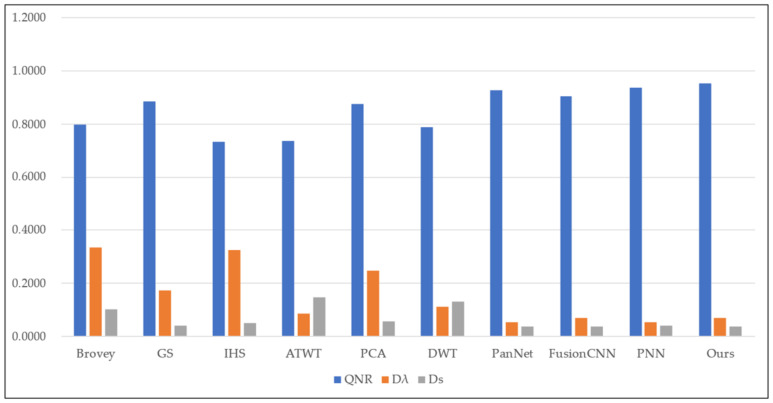
Histogram result of the third group of evaluation indexes.

Figure 15 shows the fusion results of different fusion methods on the last set of remote sensing images. Because one of the two football fields in this geographical location is a real turf and the other is artificial turf, there are some differences between the two football fields in the input source images. Figure 15c,h retain better spatial resolution in the building area, but have more severe spectral distortion, obtaining too dark and too bright spectra, respectively. Figure 15i,k present the same problems, The $D_{\lambda }$ index in Figure 15i also reached the best value, but its spatial resolution was very low, and the overall image appeared blurred. Figure 15j has a higher spatial resolution but still has some shortcomings in terms of spectral preservation compared with our method (Figure 15i). In terms of subjective visual effects, our method outperforms the other algorithms in terms of spectral preservation and texture detail. Table 5 and Figure 16 present the evaluation indexes for the fourth set of data, where the bold numbers indicate the best value for each evaluation index. Although our method does not obtain the best $D_{\lambda }$ metric, combined with the subjective visual results in Figure 15, our proposed algorithm outperforms the other fusion methods overall, especially in terms of spatial resolution. 

**Figure 15 sensors-22-07339-f015:**
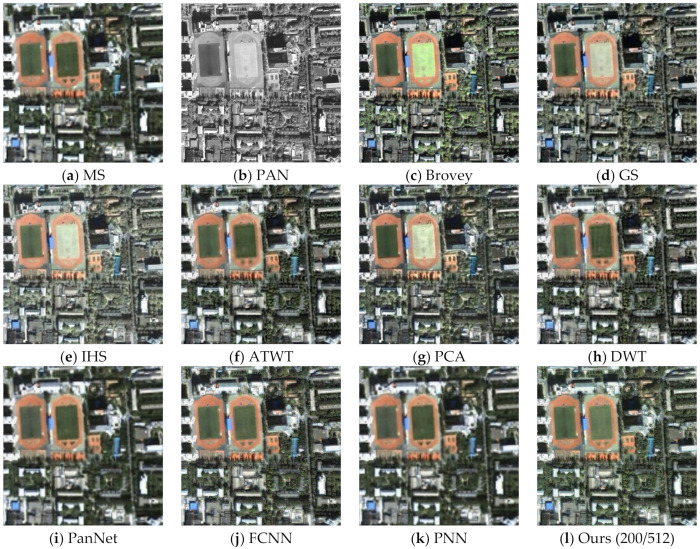
Fusion results for the fourth group of remote sensing image data.

**Figure 16 sensors-22-07339-f016:**
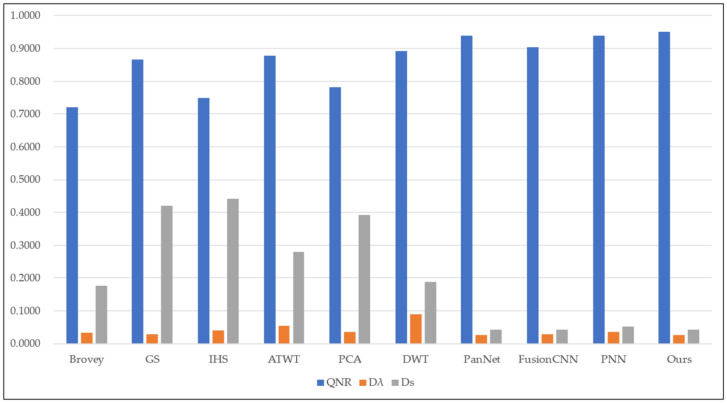
Histogram result of the fourth group of evaluation indexes.

## 5. Conclusions

In this paper, we propose a remote sensing image fusion method using morphological convolutional neural networks with information entropy for optimal scale. Our method extracts the texture and cartoon components of remote sensing images at multi-scale using MCA and selects the best scale using information entropy theory. The spectral and spatial information of the input image is fully utilized while avoiding information loss. In the network design stage, we obtain the final fusion result using an attentional convolutional neural network to retain source image information while enhancing the extraction of the input image details. We provide an experimental analysis on different types of data acquired from different satellites to demonstrate that our method better maintains the spectral information and obtains richer spatial details than existing fusion methods.

In future work, we will keep using the idea of MCA combined with deep learning to apply this work not only to MS image and PAN image fusion. Our scheme can be improved by continuing to refine the network structure to apply hyperspectral image and MS image fusion or hyperspectral image and PAN image fusion.

## Figures and Tables

**Figure 1 sensors-22-07339-f001:**
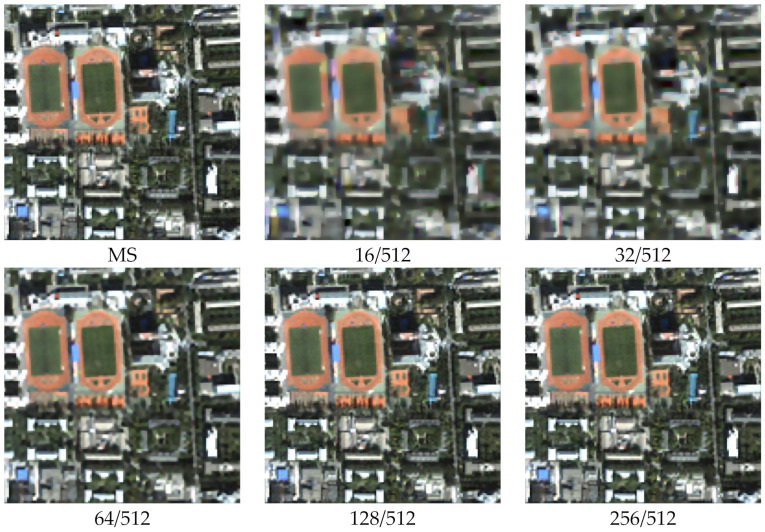
Cartoon components of the MS image at different scales.

**Figure 2 sensors-22-07339-f002:**
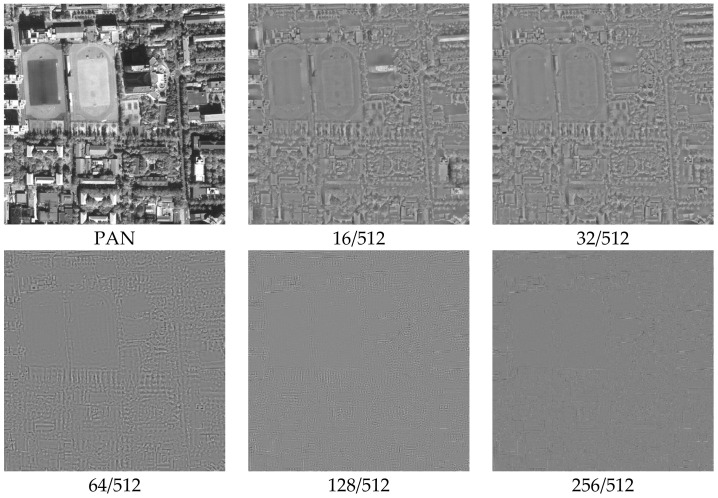
Cartoon components of the PAN image at different scales.

**Figure 3 sensors-22-07339-f003:**
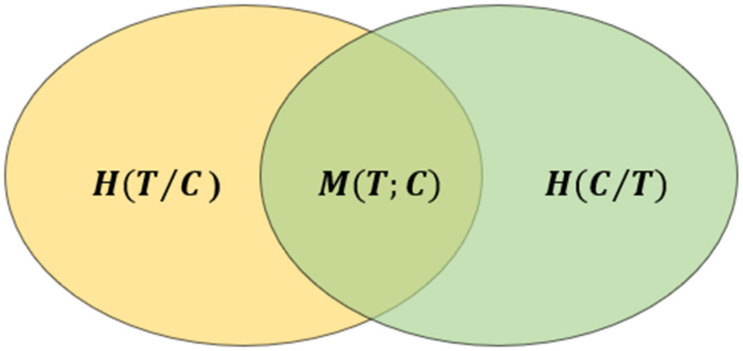
The relationship of information entropy between images *T* and *C*.

**Figure 4 sensors-22-07339-f004:**
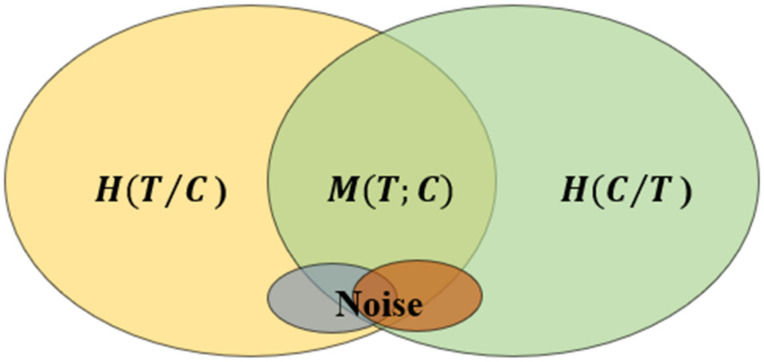
The relationship between noised image *T* and noised image *C*.

**Figure 5 sensors-22-07339-f005:**
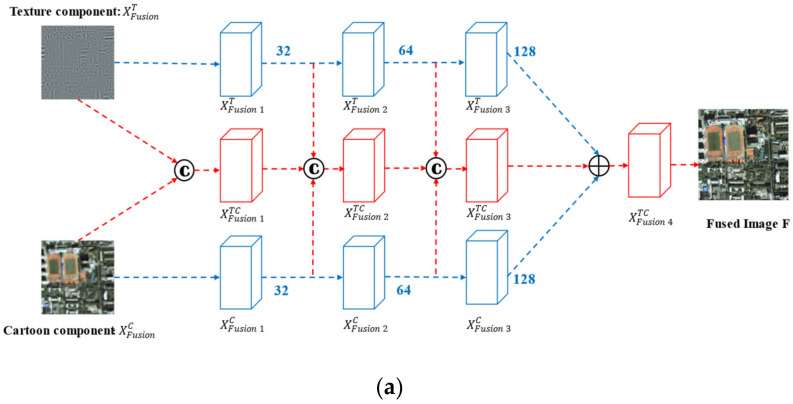
The overall frame diagram of the fusion network and structure details. (**a**) The overall frame diagram of the fusion network; (**b**) The structure details of the network.

**Figure 6 sensors-22-07339-f006:**
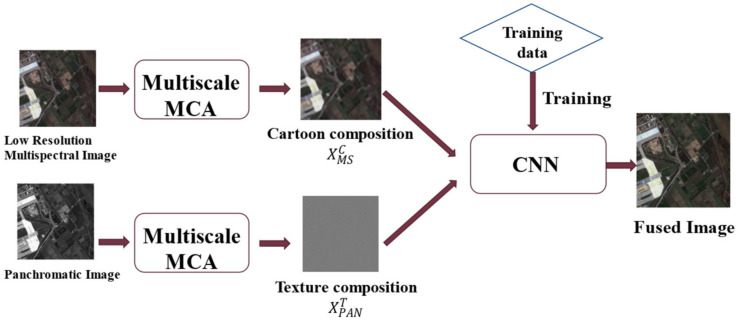
The flow chart of the fusion algorithm.

**Figure 7 sensors-22-07339-f007:**
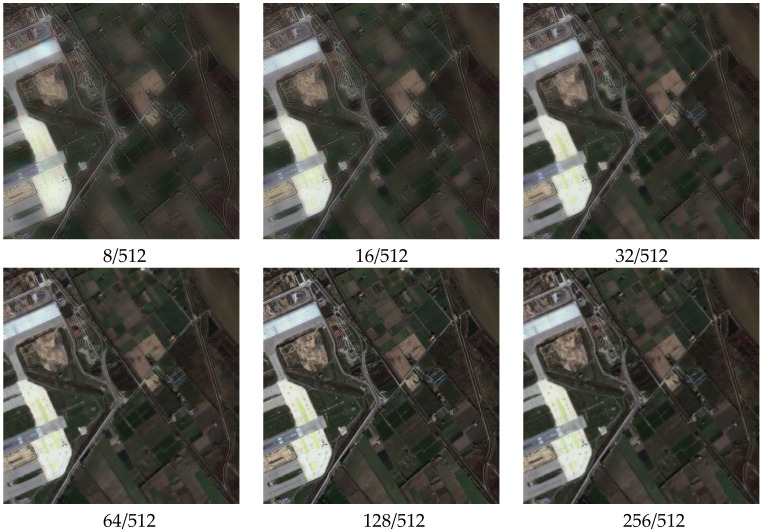
Fusion results at different scales.

**Figure 8 sensors-22-07339-f008:**
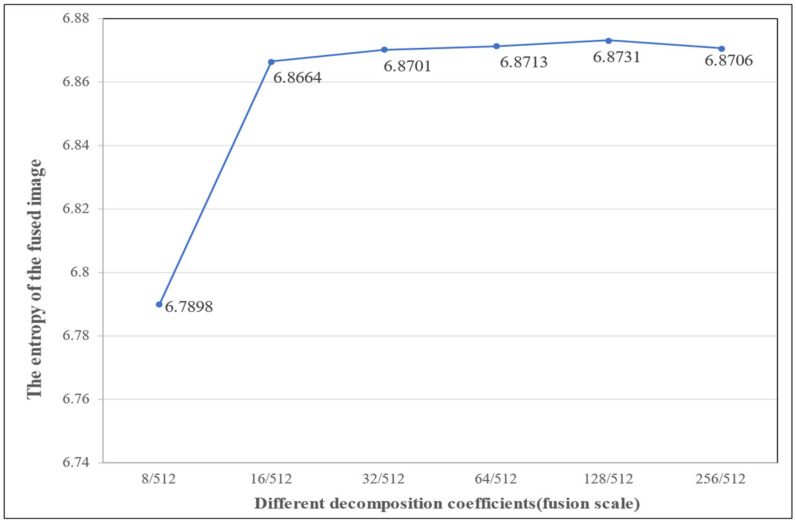
Information entropy line chart of fused images.

**Table 1 sensors-22-07339-t001:** Information entropy of fused images at different scales.

Different decomposition coefficients (fusion scale)	8/512	16/512	32/512	64/512	128/512	256/512
The entropy of the fused image	6.7898	6.8664	6.8701	6.8713	**6.8731**	6.8706

**Table 2 sensors-22-07339-t002:** The first group evaluation indexes of different fusion results.

Fusion Method	PSNR	ERGAS	RMSE	CC
Brovey	26.4100	6.1519	20.1044	0.9806
GS	28.7420	5.1143	20.0054	0.9811
IHS	28.7369	5.1160	20.0105	0.9810
ATWT	28.8015	6.8793	19.9130	0.9740
PCA	28.5646	6.0954	20.1777	0.9790
DWT	27.4118	6.9636	19.1508	0.9726
PanNet	28.5425	5.1880	18.2084	0.9908
FusionCNN	27.5199	4.3925	17.7332	0.9916
PNN	27.9194	4.4152	18.8455	0.9815
Ours	**28.8136**	**3.8734**	**16.2711**	**0.9934**

**Table 3 sensors-22-07339-t003:** The second group evaluation indices of different fusion results.

Fusion Method	PSNR	ERGAS	RMSE	CC
Brovey	31.9912	5.4425	27.3828	0.8618
GS	32.5129	5.2646	26.6783	0.8663
IHS	32.4823	5.2995	26.7186	0.8664
ATWT	35.8258	2.5560	12.9034	0.9648
PCA	31.9379	5.4145	27.4569	0.9657
DWT	34.3709	2.6027	13.1378	0.9635
PanNet	37.4864	3.1304	14.7722	0.9653
FusionCNN	37.6786	2.5543	12.8365	0.9804
PNN	37.5268	3.1121	14.8292	0.9671
Ours	**38.3922**	**2.5403**	**12.7677**	**0.9809**

**Table 4 sensors-22-07339-t004:** The third group evaluation indexes of different fusion results.

Fusion Method	QNR	$D_{\lambda }$	$D_{s}$
Brovey	0.7979	0.3343	0.1022
GS	0.8863	0.1717	0.0407
IHS	0.7316	0.3245	0.0503
ATWT	0.7375	0.0843	0.1465
PCA	0.8767	0.2464	0.0562
DWT	0.7886	0.1114	0.1302
PanNet	0.9273	**0.0522**	0.0373
FusionCNN	0.9041	0.0679	0.0356
PNN	0.9368	0.0530	0.0396
Ours	**0.9528**	0.0696	**0.0348**

**Table 5 sensors-22-07339-t005:** The fourth group evaluation indexes of different fusion results.

Fusion Method	QNR	$D_{\lambda }$	$D_{s}$
Brovey	0.7200	0.0323	0.1754
GS	0.8673	0.0278	0.4198
IHS	0.7481	0.0399	0.4409
ATWT	0.8777	0.0531	0.2787
PCA	0.7813	0.0365	0.3911
DWT	0.8917	0.0896	0.1870
PanNet	0.9388	**0.0253**	0.0424
FusionCNN	0.9041	0.0273	0.0427
PNN	0.9385	0.0343	0.0528
Ours	**0.9515**	0.0262	**0.0418**

## Data Availability

Not applicable.

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
