# Peer review of "Remote Sensing Image Fusion Based on Morphological Convolutional Neural Networks with Information Entropy for Optimal Scale"

_sensors, 2022, doi:10.3390/s22197339_

Round 1
Reviewer 1 Report
Abstract: good. no comments.
Introduction: It can be improved. What is the gap? what is the issue? What is your suggestion to improve? Also, more relevant new (recent two years) references should be used in the introduction.
Besides, the references are old. It is necessary to use more new and recent references.
Method:
Please add ''Methods'' after the introduction and before your section.
I did not understand the line chart of ''the information entropy of the fused image (figure 8)''. Please explain more about the method. How the optimal fusion result is obtained by using the parameter 128/512?
Results and discussion
Please add ''Results and discussion'' as the main section name.
I think the "Evaluation indexes" needs more potent discussion using new recent references.
For the sentence" a larger correlation parameter indicates more similarity between two images".. do you have any reference? And also for the sentences of this paragraph.
Conclusion
''better maintains the 398 spectral information and obtains richer spatial details than existing fusion methods.'' How? Please mention it briefly in your conclusion.
plagiarism:
I checked it. The overall percentage is low. (15). However, in some sections the similarity is a little high. I attached the similarity detection file. Please check. Thanks

Reviewer 2 Report
1. Please polish the Abstract. Please add sentences to explain the meaning, the main points, the improvement and the promising application of the study. Description on the major work should be should be shortened.
2. Please improve the logic of Introduction. It should declare the background, progress and motivation of this manuscript.
3. It is strongly suggested that the comment on the published works should be accurately described. The potential meaning of the study should be carefully considered in Introduction. The logic in Introduction should be checked and improved.
4. The last two paragraphs in Introduction should be combined.
5. Please refine the subtitle. It should be simple and accurate. i.e., title of Section 2 should be improved.
6. Please explain why a “-6.8701” exists in Fig.8.
7. Please clearly describe the results: what has been obtained from the study and the potential use of the study. Please revise the conclusion.
Reviewer 3 Report
please see Attachment

Round 2
Reviewer 2 Report
Please try to improve the English expressions in the whole manuscript.
The first sentence in Abstract should be corrected. Please check the similar problems in the whole manuscript.
Author Response
Thank you for your suggestion. We have re-checked the whole manuscript and asked an native English expert to polish it. We hope this version will be satisfying.
Reviewer 3 Report
please see Attachment

Author Response
On behalf of my co-authors, we appreciate the reviewers and editors very much for their contribution and help on our manuscript.